# Tackling antibiotic resistance by inducing transient and robust collateral sensitivity

Sara Hernando-Amado [1] ✉, Pablo Laborda [1,2,3] & José Luis Martínez [1] ✉

Collateral sensitivity (CS) is an evolutionary trade-off traditionally linked to the mutational acquisition of antibiotic resistance (AR). However, AR can be temporally induced, and the possibility that this causes transient, non-inherited CS, has not been addressed. Mutational acquisition of ciprofloxacin resistance leads to robust CS to tobramycin in pre-existing antibiotic-resistant mutants of *Pseudomonas aeruginosa*. Further, the strength of this phenotype is higher when *nfxB* mutants, over-producing the efflux pump MexCD-OprJ, are selected. Here, we induce transient *nfxB*-mediated ciprofloxacin resistance by using the antiseptic dequalinium chloride. Notably, non-inherited induction of AR renders transient tobramycin CS in the analyzed antibiotic-resistant mutants and clinical isolates, including tobramycin-resistant isolates. Further, by combining tobramycin with dequalinium chloride we drive these strains to extinction. Our results support that transient CS could allow the design of new evolutionary strategies to tackle antibiotic-resistant infections, avoiding the acquisition of AR mutations on which inherited CS depends.

Antibiotic resistance (AR) of bacterial pathogens constitutes a major threat to human health. Among bacteria causing the greatest concern, *Pseudomonas aeruginosa* stands out[1,2]. This bacterium is an opportunistic pathogen causative of nosocomial infections, as well as of chronic infections in patients suffering from chronic obstructive pulmonary disease or cystic fibrosis[3–5]. *P. aeruginosa* presents a low intrinsic susceptibility to different antibiotics[6,7] and, in addition, it can increase its level of resistance through the acquisition of mutational changes[8,9]. Among the different AR determinants of *P. aeruginosa*, multidrug efflux pumps should be highlighted, since they contribute to intrinsic, acquired, and transient AR[10–13].

Expression of efflux pumps encoding genes is usually low under regular growing conditions. However, antibiotic-resistant isolates overexpressing them, as the consequence of mutations in genes encoding their regulators, are frequently isolated from patients under treatment. In addition, the expression of multidrug efflux pumps encoding genes can be transiently triggered in the presence of specific effectors that bind said regulators[14–19]. In the case of MexCD-OprJ of *P. aeruginosa*, the operon encoding this efflux pump is overexpressed when loss-of-function mutations in the gene that encodes its local

repressor, *nfxB*[20], are acquired. In addition, *mexCD-oprJ* is over-expressed when inducer compounds, such as benzalkonium chloride, dequalinium chloride, tetraphenylphosphonium chloride, chlorhexidine, ethidium bromide, rhodamine 6G, as well as the antimicrobial human peptides LL-37, procaine or atropine, temporally induce its expression, possibly by binding NfxB[13,21–24].

MexCD-OprJ overproduction provides resistance to tetracyclines, chloramphenicol, and quinolones (i.e., ciprofloxacin)[25,26]. Hence, the presence of inducers might compromise the efficacy of the antibiotics that are substrates of this efflux pump. In particular, the potential risk associated with the utilization of antiseptics, disinfectants, and anesthetics, commonly used in hospitals, due to the transient induction of ciprofloxacin resistance in *P. aeruginosa* they render, has been highlighted[13]. However, it is also well known that *nfxB* loss-of-function mutants over-expressing *mexCD-oprJ* present an inherited collateral sensitivity (CS) to aminoglycosides (i.e., tobramycin) and β-lactams (i.e., aztreonam)[20,27,28]. It has been proposed that the molecular cause behind CS to aminoglycosides in *mexCD-oprJ*-overexpressing mutants is a reduction of the amount of the outer membrane protein OprM, which forms part of the aminoglycosides' efflux pump MexXY[20]. Since

[1]Centro Nacional de Biotecnología, CSIC, 28049 Madrid, Spain. [2]The Novo Nordisk Foundation Center for Biosustainability, Technical University of Denmark, 2800 Kgs. Lyngby, Denmark. [3]Department of Clinical Microbiology 9301, Rigshospitalet, 2100 Copenhagen, Denmark. ✉e-mail: shernando@cnb.csic.es; jlmtnez@cnb.csic.es

CS to aminoglycosides robustly emerges after ciprofloxacin exposure, it has been proposed that this phenotype could be exploitable for the improvement of treatments against *P. aeruginosa* infections[20,27–29]. Therefore, here we analyze if it would be possible to temporarily induce and exploit transient CS to tobramycin associated with the transient overproduction of MexCD-OprJ.

CS is a trade-off of AR evolution by which the acquisition of resistance to an antibiotic leads to an increased susceptibility to another[30,31]. While some cases of CS associated with the acquisition of AR genes by horizontal gene transfer have been described[32], most works in the field have focused on mutational resistance[30,31,33,34]. Different adaptive laboratory evolution (ALE) assays have been performed to analyze the possible combination[35–37] or alternation of drug pairs[27,38–40]. However, the fact that different patterns of CS emerge in replicate populations of a single genetic background submitted to ALE in presence of the same drug[41–43], complicates the applicability of this evolutionary trade-off. Further, different isolates of *P. aeruginosa*, including antibiotic-resistant mutants, may be present in cystic fibrosis patients after exposure to antimicrobial therapies[8,9,44]; each one could present a different CS pattern when challenged with the same drug[45]. This is because pleiotropic and epistatic phenomena determine fitness costs associated with the acquisition of AR and therefore, the possible mutations—and the associated trade-offs—that can be selected in a specific genetic background[46–49]. Therefore, the existence of different genetic backgrounds limits the clinical exploitation of CS to those cases in which a robust pattern of CS emerges when a drug is used. This conservation may occur due to parallel evolution[50], although resistance to a specific antibiotic in different genetic backgrounds more often occurs by the acquisition of different genetic variations. Therefore, it would be unlikely—although very desirable—that all of them present the same pattern of CS.

In this regard, we have recently identified a robust tobramycin CS associated with the acquisition of different ciprofloxacin resistance mutations in varied pre-existing antibiotic-resistant mutants of *P. aeruginosa*[28]. The same robust CS pattern was also found in clinical isolates of *P. aeruginosa* presenting different genomic backgrounds and mutational resistomes[29]. The mutations involved in this phenotype were acquired in ciprofloxacin target encoding genes *gyrAB* (encoding the DNA gyrase)[51–55] and in *nfxB* or *mexS* genes (encoding negative regulators of the expression of *mexCD-oprJ* and *mexEF-oprN*, respectively, which encode efflux pumps that extrude ciprofloxacin)[26,53,56–58]. Notably, the strength of CS to tobramycin, an antibiotic that, as ciprofloxacin, forms part of usual therapies against *P. aeruginosa* infections[59], was higher in the genetic backgrounds that acquired mutations in *nfxB*[28]. In fact, it has been suggested that the phenotypic convergence towards CS to aminoglycosides observed in clinical isolates of *P. aeruginosa* may be mainly associated with the acquisition of genetic variations in *nfxB*[27,58,60].

The exploitation of CS, as explored so far, requires the selection of antibiotic-resistant mutants to a first drug[30]. A seminal work has identified promising combination therapies making use of antimicrobial peptides, as they enhance the activity of other antibiotics and slow down the de novo evolution of resistance[61]. However, selection of resistant mutants—even presenting CS to a second drug—carries a risk if these mutants are not efficiently eradicated by the second antibiotic of the pair. This risk could be eliminated by temporarily inactivating the proteins encoded by those genes whose mutations lead to acquired AR and CS. The reason is that, in this situation, selection of resistant mutants would not be a requirement to achieve CS (Conceptual Fig. 1).

In this work we transiently force the mentioned *nfxB*-mediated mechanism of ciprofloxacin resistance in our set of PA14 isogenic mutants and in a set of *P. aeruginosa* clinical strains, using an inducer of the expression of *mexCD-oprJ*[13]. Among the possible inducers we choose dequalinium chloride (DC), since it presents many pharmacological and therapeutic benefits, including its antiseptic and disinfectant capacities[62–64], its anticancer properties[65–69], as well as its antiparasitic and antiviral activities[70,71]. We test whether a temporary and robust CS to tobramycin could emerge in different *P. aeruginosa* strains, including antibiotic-resistant mutants, when DC was present.

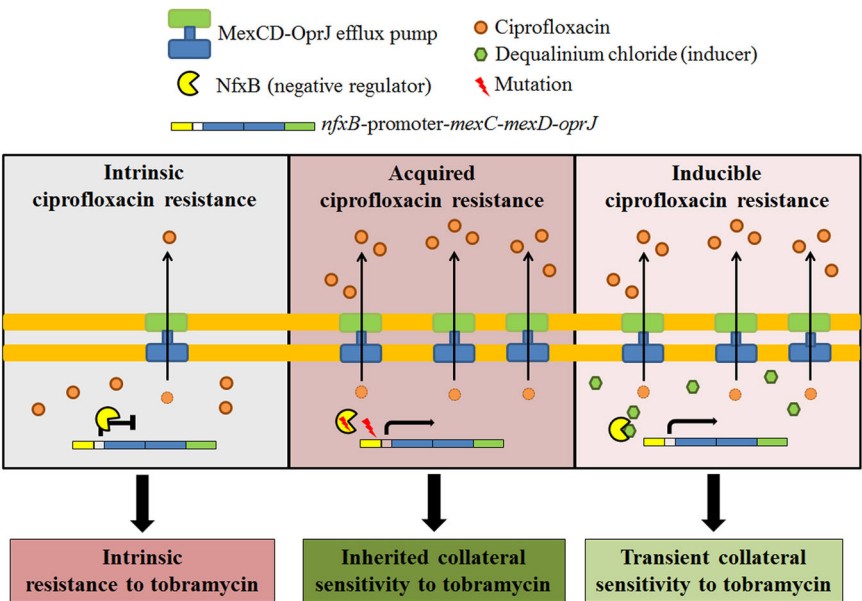

**Fig. 1 | Conceptual figure showing acquired or inducible collateral sensitivity to tobramycin associated with stable or transient over-production of the efflux pump MexCD-OprJ.** Expression of MexCD-OprJ efflux pump encoding genes is low under regular growing conditions. However antibiotic-resistant isolates overproducing it, due to the acquisition of loss-of-function mutations in the gene that encodes its negative regulator, *nfxB*, are selected in the presence of ciprofloxacin, presenting inherited resistance to ciprofloxacin and collateral sensitivity to tobramycin. In addition, transient resistance to ciprofloxacin can be temporally induced by different compounds, such as dequalinium chloride, that cause a temporal—non-inherited—overexpression of *mexCD-oprJ*. In this situation, we hypothesize that transient collateral sensitivity to tobramycin could emerge and that it could be possible exploiting it, reducing the risk of selecting antibiotic-resistant mutants which precedes the acquisition of stable CS.

Our findings show the existence of transient CS to antibiotics as the result of non-inherited AR, which is not associated with the fixation of genetic variations. In addition, we demonstrate that transient CS may drive antibiotic-resistant mutants and clinical strains of *P. aeruginosa* to extinction.

## Results

### Dequalinium chloride induces a robust and transient tobramycin hyper-susceptible state in antibiotic-resistant mutants of *P. aeruginosa*

Using a set of pre-existing antibiotic-resistant mutants of *P. aeruginosa* PA14 containing single (*nfxB177*, *parR87*, *orfN50*, *mexZ43*) or multiple mutations (MDR6 and MDR12) (Supplementary Table 1), we recently described that the mutational acquisition of ciprofloxacin resistance renders a robust CS to tobramycin[28]. In particular, *mexS* variants were selected in the model strain PA14 and in *mexZ43*, *parR87*, and MDR6 mutants, *gyrAB* variants were selected in *nfxB177* and MDR12 mutants, and *nfxB* variants were selected in *orfN50*, *mexZ43*, *parR87*, and MDR12 mutants. Importantly, genetic variations in *nfxB* led to the strongest CS to tobramycin. Unfortunately, mutations in this gene are not the ones more likely selected in some of the analyzed genetic backgrounds, due to historical contingency. This means that, under ciprofloxacin selective pressure, some of the selected mutants will become less susceptible to tobramycin than they would have been if they had acquired mutations in *nfxB*. Therefore, we decided to avoid the effect of genetic background in shaping the evolution of ciprofloxacin resistance (and of CS to tobramycin) by specifically inducing a *nfxB*-mediated ciprofloxacin resistance transient status[13] in the mentioned set of mutants, using the antiseptic DC.

To analyze the capacity of DC to induce CS to tobramycin and confirm its capacity to induce resistance to ciprofloxacin, as previously described[13], we determined the ciprofloxacin and tobramycin minimal inhibitory concentrations (MICs) in the presence or absence of this compound, in the seven genetic backgrounds. MIC variations (determined using MIC test strips, that allow the discrimination of small MIC changes) above or below an increase or a decrease of 2-fold, respectively, indicated biologically relevant variations of resistance, as previously described[28,72]. As expected, DC caused a transient induction of ciprofloxacin resistance in all the genetic backgrounds analyzed, with exception of the already ciprofloxacin-resistant mutant *nfxB177*, which stably overexpresses *mexCD-oprJ* (Fig. 2; Table 1). In particular, ciprofloxacin MIC increased 11.7-fold in PA14, 8-fold in *orfN50*, 5.3-fold in *parR87*, 2.6-fold in *mexZ43* and in MDR12 or 2-fold in MDR6. Interestingly, resistance levels were considerably below the ones previously observed after ALE in the presence of ciprofloxacin[28] (Fig. 3): up to

128-fold in *parR87*, 48-fold in *mexZ43*, 46.9-fold in PA14, 24-fold in MDR6, 16-fold in *nfxB177*, 15.8-fold in *orfN50*, or 10.5-fold in MDR12. This indicates that the use of DC is less risky than that of ciprofloxacin since it does not only reduce the levels of ciprofloxacin resistance that are acquired, but AR is also transient and non-inheritable.

Then, we tested whether transient induction of *nfxB*-mediated ciprofloxacin resistance could lead to transient CS to tobramycin (Conceptual Fig. 1). In agreement with our hypothesis, DC caused a transient induction of CS to tobramycin in all the genetic backgrounds analyzed with exception of the aforementioned loss-of-function mutant *nfxB177* (Fig. 2; Table 1). In particular, tobramycin MIC decreased up to 6-fold in MDR12, 4-fold in *parR87* and in *orfN50*, 2.6-fold in *mexZ43*, and in MDR6 or 2-fold in PA14. It is important to mention that 2 out of the 7 analyzed genetic backgrounds were originally resistant to tobramycin (MIC higher than 2 μg/ml) and 3 genetic backgrounds were close to resistant to tobramycin (MIC of 2 μg/ml), following the criteria of the European Committee on Antimicrobial Susceptibility Testing. Moreover, we observed a decrease of tobramycin MICs similar, or even higher, than those selected in ALE assays in the presence of ciprofloxacin[28] (Fig. 3): a decrease up to 6-fold in *orfN50*, 3-fold in *parR87* and MDR6, 2.6-fold in PA14 or 2-fold in *nfxB177*, *mexZ43* and MDR12. This suggests that it is not only possible to induce CS to tobramycin by using DC in all the genetic backgrounds analyzed (with exception of the mutant *nfxB177*) but that the strength of this phenotype is improved in some genetic backgrounds, such as *mexZ43*, *parR87*, or MDR12, as compared with the one observed in ciprofloxacin-resistant mutants. It is also important to highlight that while the acquisition of CS to tobramycin associated with the use of ciprofloxacin depends on the acquisition of high-level stable, inherited, ciprofloxacin resistance, induction of CS to tobramycin associated with the use of DC depends on more moderate levels of transient, non-inherited, ciprofloxacin resistance. These results also indicate that, while stable CS to tobramycin in some mutational backgrounds would be linked to important levels of stable, inherited, resistance to ciprofloxacin, depending on the possible mutations they may acquire, the induction of transient and fairly mild *nfxB*-mediated resistance to ciprofloxacin in all of them renders an improved CS to tobramycin (Fig. 3).

### Dequalinium chloride induction of CS is robust and specific for aminoglycosides

It has been described that several pharmacological properties and therapeutic benefits of DC are due to its capacity to interfere with the activity of diverse proteins[73]. Therefore, to analyze whether the effect of DC on susceptibility to tobramycin is specific (and also occurs for

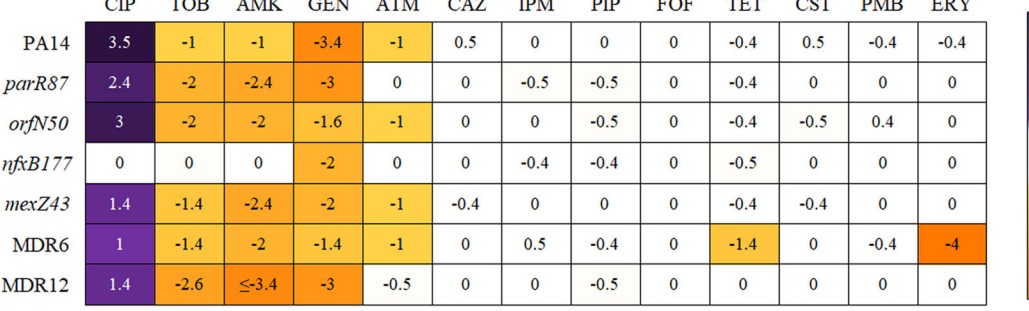

| | CIP | TOB | AMK | GEN | ATM | CAZ | IPM | PIP | FOF | TET | CST | PMB | ERY |
|---|---|---|---|---|---|---|---|---|---|---|---|---|---|
| PA14 | 3.5 | -1 | -1 | -3.4 | -1 | 0.5 | 0 | 0 | 0 | -0.4 | 0.5 | -0.4 | -0.4 |
| *parR87* | 2.4 | -2 | -2.4 | -3 | 0 | 0 | -0.5 | -0.5 | 0 | -0.4 | 0 | 0 | 0 |
| *orfN50* | 3 | -2 | -2 | -1.6 | -1 | 0 | 0 | -0.5 | 0 | -0.4 | -0.5 | 0.4 | 0 |
| *nfxB177* | 0 | 0 | 0 | -2 | 0 | 0 | -0.4 | -0.4 | 0 | -0.5 | 0 | 0 | 0 |
| *mexZ43* | 1.4 | -1.4 | -2.4 | -2 | -1 | -0.4 | 0 | 0 | 0 | -0.4 | -0.4 | 0 | 0 |
| MDR6 | 1 | -1.4 | -2 | -1.4 | -1 | 0 | 0.5 | -0.4 | 0 | -1.4 | 0 | -0.4 | -4 |
| MDR12 | 1.4 | -2.6 | ≤-3.4 | -3 | -0.5 | 0 | 0 | -0.5 | 0 | 0 | 0 | 0 | 0 |

Log₂(Fold change MIC) — scale 4 to -4

**Fig. 2 | Robustness of collateral sensitivity to aminoglycosides in PA14 and different resistant mutants in the presence of dequalinium chloride.** Susceptibility to antibiotics from different structural families was analyzed in PA14 wild-type strain and in the genetic backgrounds *parR87*, *orfN50*, *nfxB177*, *mexZ43*, MDR6, and MDR12 in the presence or absence of DC. Intensity of the color is proportional to the log-transformed fold change of MIC in the presence of DC regarding the MIC of the respective parental strain in the absence of DC and values are indicated in the figure. Changes of MICs above or below an increase or a decrease of 2-fold, respectively, were considered biologically relevant to classify a population as "resistant" (purple) or "susceptible" (yellow), as previously described[28]. MIC values (μg/ml) are included in Table 1. CIP: ciprofloxacin, TOB: tobramycin, AMK: amikacin, GEN: gentamycin, ATM: aztreonam, CAZ: ceftazidime, IPM: imipenem, PIP: piperacillin, FOF: fosfomycin, TET: tetracycline, CST: colistin, PMB: polimixin B, ERY: erythromycin.

**Table 1 | MIC values (µg/ml) of antibiotics from different structural families in absence (−) or presence (+) of 10 µg/ml of dequalinium chloride in PA14 and different *P. aeruginosa* antibiotic-resistant mutants**

| | | CIP | TOB | AMK | GEN | ATM | CAZ | IPM | PIP | FOF | TET | CST | PMB | ERY |
|---|---|---|---|---|---|---|---|---|---|---|---|---|---|---|
| −DC | PA14 | 0.064 | 1 | 2 | 2 | 3 | 1 | 1 | 2 | 48 | 16 | 2 | 2 | 256 |
| | *parR87* | 0.19 | 2 | 8 | 3 | 2 | 1 | 1.5 | 3 | 48 | 16 | 2 | 1.5 | >256 |
| | *orfN50* | 0.125 | 3 | 6 | 3 | 8 | 3 | 1.5 | 6 | 8 | 32 | 3 | 1.5 | >256 |
| | *nfxB177* | 2 | 0.5 | 1 | 0.5 | 1.5 | 1.5 | 0.25 | 2 | 24 | 48 | 2 | 2 | >256 |
| | *mexZ43* | 0.19 | 2 | 4 | 2 | 3 | 1 | 0.75 | 2 | 32 | 16 | 2 | 2 | >256 |
| | MDR6 | 0.125 | 2 | 8 | 2 | 1.5 | 0.75 | 0.5 | 2 | 24 | 32 | 6 | 4 | 128 |
| | MDR12 | 0.38 | 48 | >256 | 64 | 1.5 | 0.5 | 0.75 | 3 | 32 | 16 | 8 | 12 | >256 |
| +DC | PA14 | 0.75 | 0.5 | 1 | 0.19 | 1.5 | 1.5 | 1 | 2 | 48 | 12 | 3 | 1.5 | 192 |
| | *parR87* | 1 | 0.5 | 1.5 | 0.38 | 2 | 1 | 1 | 2 | 48 | 12 | 2 | 1.5 | >256 |
| | *orfN50* | 1 | 0.75 | 1.5 | 1 | 4 | 3 | 1.5 | 4 | 8 | 24 | 2 | 2 | >256 |
| | *nfxB177* | 2 | 0.5 | 1 | 0.125 | 1.5 | 1.5 | 0.19 | 1.5 | 24 | 32 | 2 | 2 | >256 |
| | *mexZ43* | 0.5 | 0.75 | 0.75 | 0.5 | 1.5 | 0.75 | 0.75 | 2 | 32 | 12 | 1.5 | 2 | >256 |
| | MDR6 | 0.25 | 0.75 | 2 | 0.75 | 0.75 | 0.75 | 0.75 | 1.5 | 24 | 12 | 6 | 3 | 8 |
| | MDR12 | 1 | 8 | 24 | 8 | 1 | 0.5 | 0.75 | 2 | 32 | 16 | 8 | 12 | >256 |

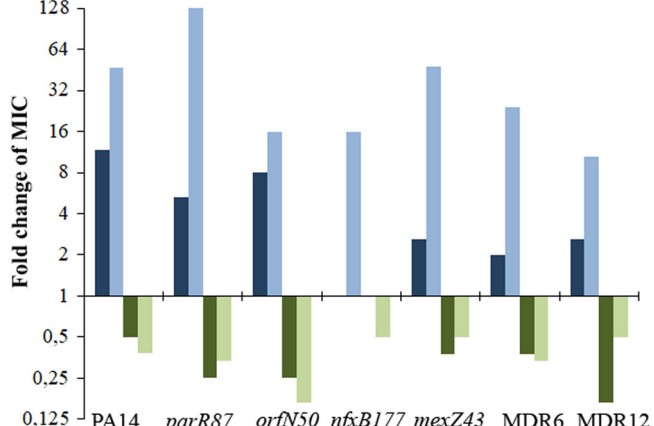

**Fig. 3 | Comparison of the effect of mutational or DC-induced acquisition of ciprofloxacin resistance on tobramycin collateral sensitivity in different isogenic *P. aeruginosa* antibiotic-resistant mutants.** We previously showed that mutational acquisition of ciprofloxacin resistance is associated with tobramycin CS[28]. The graph compares the changes in MICs of mutant strains selected in presence of ciprofloxacin with the changes in MICs of strains grown in presence of DC. Since four populations of each strain were analyzed, the figure shows data of the populations presenting largest changes in MIC. The changes in ciprofloxacin (light blue) and tobramycin (light green) MICs of PA14 wild-type strain and different resistant mutants (*parR87*, *orfN50*, *nfxB177*, *mexZ43*, MDR6, and MDR12) after ALE in the presence of ciprofloxacin respect to those ones before ALE[28] are shown. The changes in ciprofloxacin (dark blue) and tobramycin (dark green) MICs, represented as fold change, in the presence of DC respect to those in the absence of DC (due to transient ciprofloxacin and tobramycin resistance and CS, respectively) are shown. As can be seen, transient, induced ciprofloxacin and tobramycin resistance and CS, respectively, were observed for all strains except for the *nfxB177* mutant. This supports that DC induces a tobramycin CS state by principally acting on the NfxB repressor.

other aminoglycosides) or whether it can increase susceptibility to drugs from different structural families, we analyzed the MICs of a battery of antibiotics in presence or absence of this compound, in the seven aforementioned genetic backgrounds.

Firstly, we observed that DC also induces CS to other aminoglycosides, as amikacin and gentamycin (Fig. 2; Table 1). In particular, amikacin MIC decreased up to 10.6-fold in MDR12, 5.3-fold in *parR87* and *mexZ43*, 4-fold in *orfN50* and MDR6, and 2-fold in PA14, while no changes of MIC were observed in *nfxB177*. These data support that the induction caused by DC of CS to amikacin, as

previously observed for tobramycin, is *nfxB*-mediated. It is important noticing that the highest decrease of amikacin or tobramycin MICs associated with the presence of DC occurred in the genetic backgrounds MDR12 (up to 10.6- or 6-fold, respectively) and *parR87* (up to 5.3- or 4-fold, respectively), indicating the relevant role of genetic background in shaping the strength of transient CS. Gentamycin MIC decreased up to 10.5-fold in PA14, 8-fold in MDR12, 7.9-fold in *parR87*, 4-fold in *mexZ43*, 3-fold in *orfN50* and 2.6-fold in MDR6. However, gentamycin MIC decreased up to 4-fold of in *nfxB177*, suggesting that the induced CS to this aminoglycoside caused by DC in the different genetic backgrounds may be due to other causes beyond those directly related to NfxB.

Secondly, DC did not cause CS to antibiotics from other structural families (ceftazidime, imipenem, piperacillin, fosfomycin, tetracycline, erythromycin) in any of the genetic backgrounds analyzed (Fig. 2; Table 1) with the exception of the mutant MDR6, which presented increased susceptibility to both, tetracycline and erythromycin. In addition, we observed an aztreonam MIC decrease up to 2-fold in PA14, *mexZ43*, *orfN50*, and MDR6, but no changes in *parR87*, MDR12, or *nfxB177* (Fig. 2; Table 1). Since ciprofloxacin resistance mutations in *nfxB* may also cause CS to aztreonam[28], this induction of CS to aztreonam by DC in some of genetic backgrounds, and not in *nfxB177*, may also be *nfxB*-mediated by DC.

Finally, since DC might perturb the cellular permeability[73], it might be possible that this potential alteration, if it existed, impacted the susceptibility to cationic peptides, as colistin and polymyxin B. To address this possibility, we determined the MICs of those compounds in the presence and absence of DC, in the seven genetic backgrounds. No changes in MIC to these antibiotics were observed (Fig. 2; Table 1). These results support that the observed CS to tobramycin (and to other aminoglycosides) associated with the use of DC is not the result of alterations of the permeability of the bacterial membrane, at the DC concentration tested.

All together, these results indicate that the temporary *nfxB*-mediated resistance to ciprofloxacin caused by DC is also accompanied by an increased susceptibility to aminoglycosides (i.e., tobramycin) and, in some cases, to β-lactams (i.e., aztreonam), as it has been described in the case of stable, inherited, *nfxB* mutations[20,27,28].

**Evolutionary strategies based on transient and robust collateral sensitivity to drive pre-existing *P. aeruginosa* antibiotic-resistant mutants to extinction**

As shown above, we have demonstrated that it is possible to transiently produce a robust CS to tobramycin in different pre-existing antibiotic-

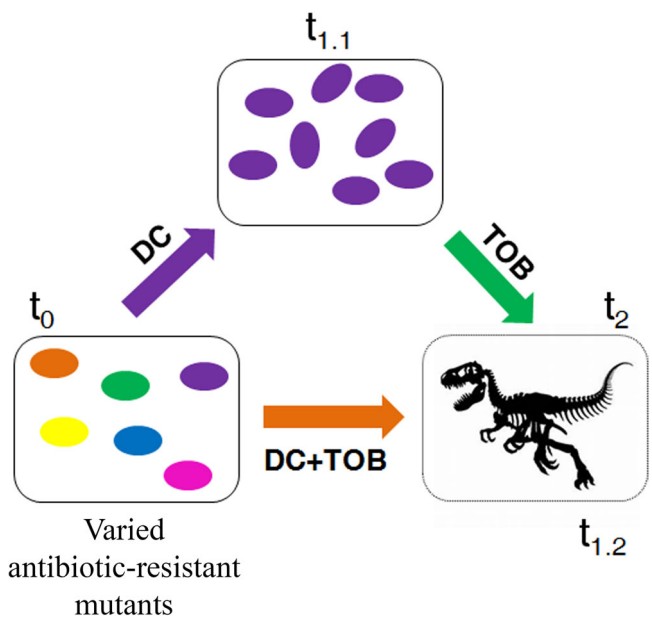

**Fig. 4 | Conceptual figure showing transient evolution of diverse antibiotic-resistant mutants of *P. aeruginosa* submitted to the alternation of dequalinium chloride with tobramycin or to the combination of dequalinium chloride with tobramycin.** Evolution starts when different antibiotic-resistant mutants are treated with dequalinium chloride (DC) or with the combination DC plus tobramycin (TOB) at time zero ($t_0$). In the first case, there is evolution towards transient, induced, ciprofloxacin resistance and transient, non-inherited, CS to tobramycin (purple cells) ($t_{1.1}$). Subsequently, treatment is switched to tobramycin that may result in the elimination (represented as a dinosaur fossil skeleton) of cells susceptible to tobramycin ($t_2$). In the second case, evolution starts when different antibiotic-resistant mutants are treated with a DC-tobramycin combination at time zero ($t_0$). Since transient ciprofloxacin resistance induced by DC leads to transient CS to tobramycin, it may be expected that drug combinations result in the elimination of cells ($t_{1.2}$), represented as a dinosaur fossil skeleton.

resistant mutants of *P. aeruginosa* PA14 (Fig. 2; Table 1). Given that we have recently described, using these mutants and clinical strains of *P. aeruginosa*, that it is possible to exploit CS to tobramycin associated with inherited resistance to ciprofloxacin[28,29] and having in mind a previous work that indicated the efficacy of aminoglycosides to eradicate quinolone-resistant populations of *P. aeruginosa* from the lungs of CF patients[27], we tested the possibility of alternating or combining DC with tobramycin to drive the mentioned pre-existing antibiotic-resistant mutants to extinction.

Our first strategy consisted in a first step on DC, to drive the analyzed mutants towards a state of transient CS to tobramycin, and a second step on tobramycin, which could drive induced tobramycin-susceptible cells to extinction (Fig. 4). Since it is known that CS may also optimize combinatory therapy[35] and we previously demonstrated that the combination of tobramycin with ciprofloxacin, the drug driving CS to the first one, is very effective in driving tobramycin-susceptible cells to extinction[28,29], our second strategy consisted in a single step in the presence of the DC-tobramycin combination (Fig. 4).

We performed ALE assays with the genetic backgrounds that showed CS to tobramycin in the presence of DC (PA14, *parR87*, *orfN50*, *mexZ43*, MDR6, and MDR12) and with *nfxB177*, which did not show DC-induced CS to tobramycin, as a control (Fig. 2; Table 1). We grew the 28 populations, four replicate populations of each genetic background, in the presence of DC during one day. At this point, we switched from DC to

tobramycin in the 28 populations (Fig. 5A) and continue growing these populations during three days (see Methods). In addition, we grew 28 control populations in the absence of any compound, 28 control populations in the presence of DC, and 28 control populations in the presence of tobramycin, during four days. As shown in Fig. 5A, only 2 out of 28 populations submitted to short-term ALE in the presence of tobramycin after DC exposure became extinct. These results indicate that exploiting transient CS to tobramycin associated with the use of DC by switching from this compound to tobramycin will probably be inefficient in driving *P. aeruginosa* to extinction, at least for the mutational backgrounds here analyzed. This result might be due to a rapid loss of induction after DC is removed. Indeed, we have previously shown that the DC-mediated induction of *mexCD-oprJ* expression, and of ciprofloxacin resistance, declines soon after the antiseptic is removed[13].

For its part, we analyzed the efficacy of the combination DC-tobramycin in the 7 original genetic backgrounds. We submitted 4 replicate populations of each different genetic background to the combination DC-tobramycin (28 populations) during three days. In parallel, we grew 28 populations in DC or tobramycin, using the same concentrations present in the drug pair, and 28 populations in the absence of any drug (84 control populations), as described in Methods. As shown in Fig. 5B, all the populations from the genetic backgrounds PA14, *parR87*, *orfN50*, *mexZ43*, MDR6, and MDR12, which presented CS to tobramycin in the presence of DC (Fig. 2; Table 1), became extinct after short-term ALE in the presence of DC-tobramycin, while *nfxB177* populations, which did not present DC-induced tobramycin CS, grew. In addition, every control population evolved in either DC or tobramycin grew, as expected. These results indicate that the pair DC-tobramycin is effective in driving the pre-existing antibiotic-resistant mutants analyzed to extinction, with exception of the *nfxB177* mutant. Further, this supports that transient *nfxB*-mediated CS to tobramycin mediated by DC is most likely responsible for the extinction of the resistant mutants.

To further confirm that there are no other molecular mechanisms, beyond transient CS, responsible for the extinction observed in the antibiotic-resistant mutants analyzed, we tested the outcome of the combined ALE assay using ceftazidime, an antibiotic for which DC does not induce CS (Fig. 2; Table 1). We submitted 4 replicate populations of each different genetic background to the combination DC-ceftazidime (28 populations) during three days. In parallel, we grew 28 populations in presence of DC or ceftazidime, using the same concentrations present in the drug pair, and 28 populations in the absence of any drug (84 control populations), as described in Methods. All the populations submitted to short-term ALE in the presence of the combination DC-ceftazidime grew, confirming that transient CS to tobramycin mediated by DC is responsible for the extinction of the antibiotic-resistant mutants analyzed.

To further analyze if the efficiency of the DC-tobramycin combination could be the result of a synergistic interaction between both compounds, we performed checkerboard analyses (Methods) with PA14 and the 6 mutational backgrounds here analyzed. We did not observe synergy (neither antagonism) between DC and tobramycin in PA14 or in the mutants *parR87*, *nfxB177*, *orfN50*, or MDR6 (FIC Index ≥0.5 and ≤4 in all cases), but some synergy was observed in the mutant *mexZ43* and MDR12, a multiple mutant containing the *mexZ43* mutation (FIC Index of 0.35 in both cases) (Supplementary Table 2). This reinforces the importance of the genetic background in shaping the interactions between pairs of drugs (i.e., tobramycin and DC), as well as transient CS (i.e., to tobramycin) associated with the use of a drug (i.e., DC) (Fig. 2). Overall, these results indicate that the high efficiency of the DC-tobramycin combination in driving most of the genetic backgrounds here analyzed to extinction is due to the capacity of DC to induce transient CS to tobramycin which, in mutants containing the *mexZ43* mutation (*mexZ43* and MDR12), might be increased by synergistic interactions.

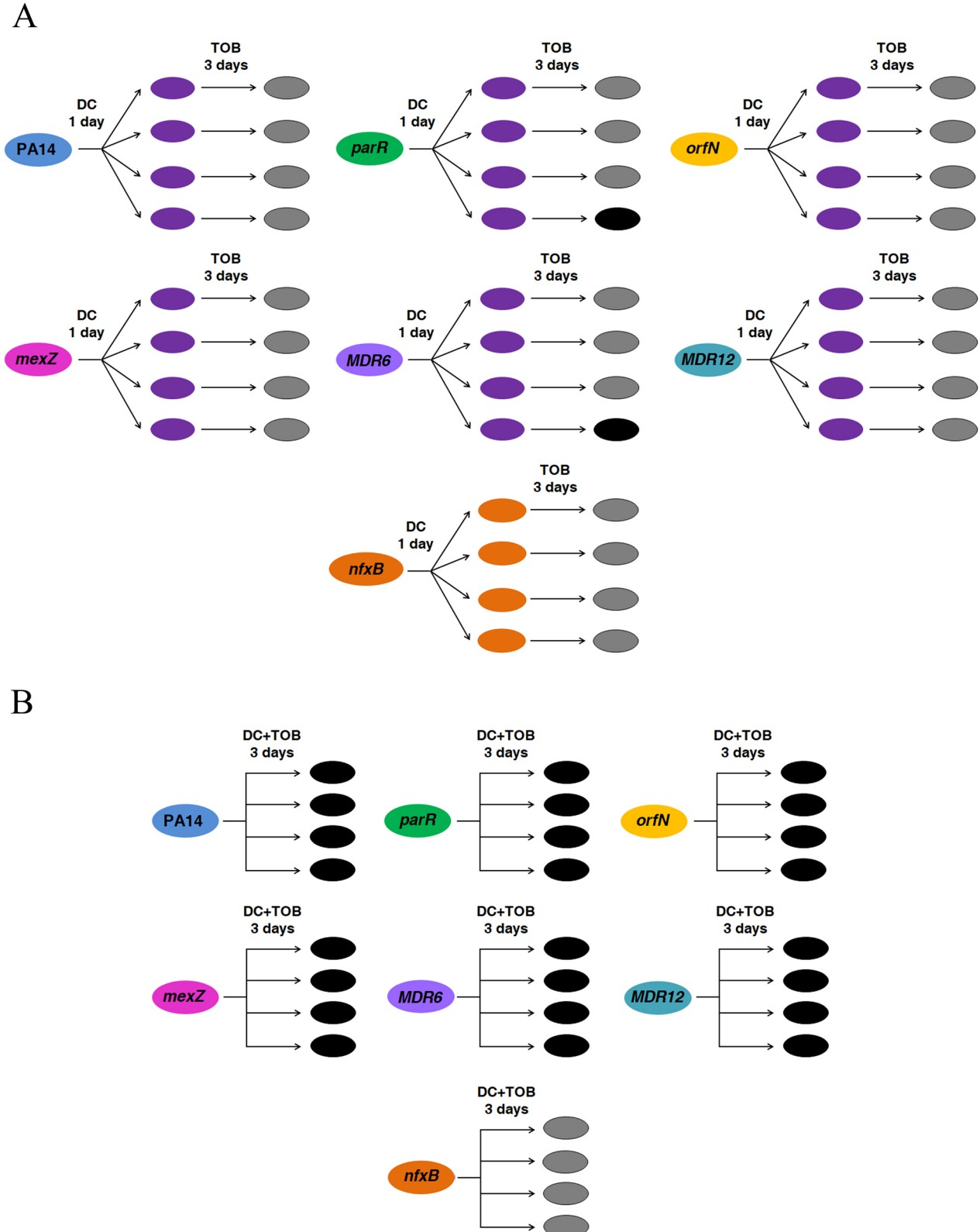

Finally, to confirm that induced resistance and CS to ciprofloxacin and tobramycin, respectively, was transient and that it could decline when the compound was removed, we analyzed MICs to ciprofloxacin and tobramycin at the end of ALE assays in populations grown during four days in presence of DC. No differences were found respect to the parental strains (Supplementary Table 3). These results indicate that DC does not select for an inherited over-production of MexCD-OprJ efflux pump, which is normally associated with mutations in *nfxB*[25,26,28]. These results are in agreement with previous findings from our laboratory in which we demonstrated a decline of the expression of *mexCD-oprJ*, previously induced by DC, two hours after the removal of this compound, to levels similar to those of non-induced cells[13]. In fact, in our previous work we sequenced *nfxB* and the promoter region of

**Fig. 5 | Diagram showing the alternation or combination of dequalinium chloride with tobramycin for driving *P. aeruginosa* PA14 and different antibiotic-resistant mutants to extinction. A** Short-term evolution of PA14 wild-type strain and 6 resistant mutants (*parR87*, *mexZ43*, *orfN50*, *nfxB177* MDR6, or MDR12), four replicate populations of each parental strain, was performed during 4 days: one day in the presence of dequalinium chloride (DC), leading to transiently ciprofloxacin-resistant and collaterally tobramycin-susceptible populations (purple cells), and three days in the presence of tobramycin (TOB). Growth of the 84 control populations was confirmed in the absence and presence of each drug during 4 days, at the concentrations present in the drugs alternation experiment. Populations extinct at the end of the ALE experiment are represented in black and surviving populations are colored in gray. Most of the populations (26 out of 28) submitted

to short-term ALE in the presence of tobramycin grew after three days. **B** Short-term evolution of PA14 wild-type strain and 6 resistant mutants (*parR87*, *mexZ43*, *nfxB177*, *orfN50*, MDR6, or MDR12), four replicate populations of each parental strain, was performed during three days in the presence of the DC-tobramycin combination. Growth of the 84 control populations was confirmed in the absence and presence of each drug, at the concentrations present in the drugs combinations. Populations extinct at the end of the ALE experiment are represented in black and surviving populations are colored in gray. All the populations that presented CS to tobramycin in the presence of DC (Fig. 2; Table 1) were extinct after short-term ALE in the presence of the combination while *nfxB177* populations grew. These results indicate that transient CS *nfxB*-mediated may improve combinatory therapy.

*mexCD-oprJ* of *P. aeruginosa* cells treated during 20 h with DC, finding no ciprofloxacin resistance mutations. To further analyze if DC could have selected other mutations, leading to cross-resistance to other antibiotics, we measured MICs to antibiotics from other structural families (ceftazidime, imipenem, aztreonam, fosfomycin, tetracycline, and erythromycin) in the populations grown during 4 days in presence of DC, finding no differences respect to the parental strains (Supplementary Table 3). Finally, and although we propose a therapeutic strategy based on a short treatment (3 days; around 20 generations), as we and others have suggested before[28,29,50,74,75], we analyzed the capacity of DC to select for ciprofloxacin or tobramycin resistance in the long term (15 days; around 100 bacterial generations). We found that this compound did not lead to an increase of ciprofloxacin or tobramycin MICs in the different genetic backgrounds of *P. aeruginosa* analyzed (Supplementary Table 4) and that, importantly, the capacity of DC to induce transient CS to tobramycin was preserved (Supplementary Table 5). Altogether, our results demonstrate that the capacity of DC to induce CS to tobramycin is temporary, rapidly reversing when its presence ceases, and that it does not select for acquired AR.

## Dequalinium chloride induces a robust and transient tobramycin-susceptible state in clinical isolates of *P. aeruginosa* which could be exploited for the design of new evolution-based therapeutic strategies

As mentioned, the combination ciprofloxacin-tobramycin is extremely effective in driving pre-existing antibiotic-resistant mutants of PA14 and clinical strains of *P. aeruginosa* presenting different genomic backgrounds, to extinction[28,29]. As demonstrated here, the combination DC-tobramycin is also able to eliminate pre-existing antibiotic-resistant mutants of PA14. However, it remained to be confirmed if the use of DC could also induce a robust and transient CS to tobramycin in clinical strains of *P. aeruginosa*, an essential requirement for the extrapolation of the previous findings to the clinical practice. Therefore, we analyzed ten clinical strains, chosen among those studied in a previous work[29], presenting different genomic backgrounds, different mutational resistomes (including tobramycin-resistant mutants) and some of them belonging to high-risk epidemic clones (ST111 and ST244) (Supplementary Table 6).

We, therefore, determined the capacity of DC to induce CS to tobramycin and confirmed its capacity to induce resistance to ciprofloxacin by measuring the ciprofloxacin and tobramycin MICs in the presence or absence of DC in ten clinical isolates of *P. aeruginosa*. As expected[13], ciprofloxacin resistance increased in the presence of DC in all the genomic backgrounds analyzed (Fig. 6; Supplementary Table 7). In particular, ciprofloxacin MICs increased from 1.9-fold up to 5.9-fold. In agreement with our observations using PA14 isogenic mutants (Fig. 3), DC caused a transient induction of CS to tobramycin in all the clinical isolates analyzed (Fig. 6; Supplementary Table 7). In particular, tobramycin MIC in presence of DC decreased from 3- up to 8-fold, being these tobramycin MIC declines even higher than the ones

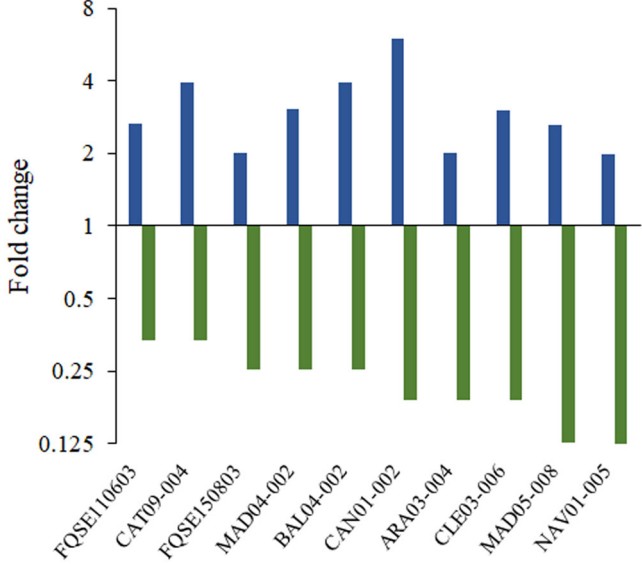

**Fig. 6 | Transient induction of ciprofloxacin resistance and collateral sensitivity to tobramycin in clinical isolates of *P. aeruginosa*.** Transient ciprofloxacin resistance (blue) and collateral sensitivity to tobramycin (green) are represented as fold change of ciprofloxacin and tobramycin MICs measured in the presence of dequalinium chloride in each clinical isolate (see Supplementary Table 7) respect to the respective MICs measured in the absence of inducer compound. As shown, dequalinium chloride induces ciprofloxacin resistance and tobramycin collateral sensitivity in all the studied clinical isolates, including originally tobramycin-resistant isolates (MAD05-008 and NAV01-005) and isolates close to be classified as resistant to this drug (CLE03-006 and ARA03-004).

observed using PA14 isogenic mutants (Fig. 3). Importantly, 4 out of the 10 analyzed clinical isolates originally resistant to tobramycin (MIC higher than 2 μg/ml) or close to resistant to this drug (MIC of 2 μg/ml) became temporally susceptible to this antibiotic (Supplementary Table 7). These results indicate that a particular CS pattern can be funneled through the induction of a specific mechanism of transient AR in a variety of *P. aeruginosa* strains, including widespread clonal complexes of clinical concern.

Therefore, we analyzed if the evolutionary strategy based on the combination of DC and tobramycin (Fig. 4, DC + TOB), that resulted in extinction of all the pre-existing antibiotic-resistant mutants of *P. aeruginosa* PA14 (Fig. 5B), could also be efficient to drive our set of ten clinical strains of *P. aeruginosa* (Supplementary Table 6) to extinction. We submitted 4 replicate populations of each different clinical isolate to the combination DC-tobramycin (40 populations). In addition, we grew 40 populations in the presence of DC or tobramycin, using the same concentrations present in the drug pair (80 control populations) and 40 populations in the absence of any drug (Methods). As shown in Fig. 7, nearly all populations (39 out of 40) submitted to short-term ALE in the presence of DC-tobramycin became extinct, while every control population grew, as expected.

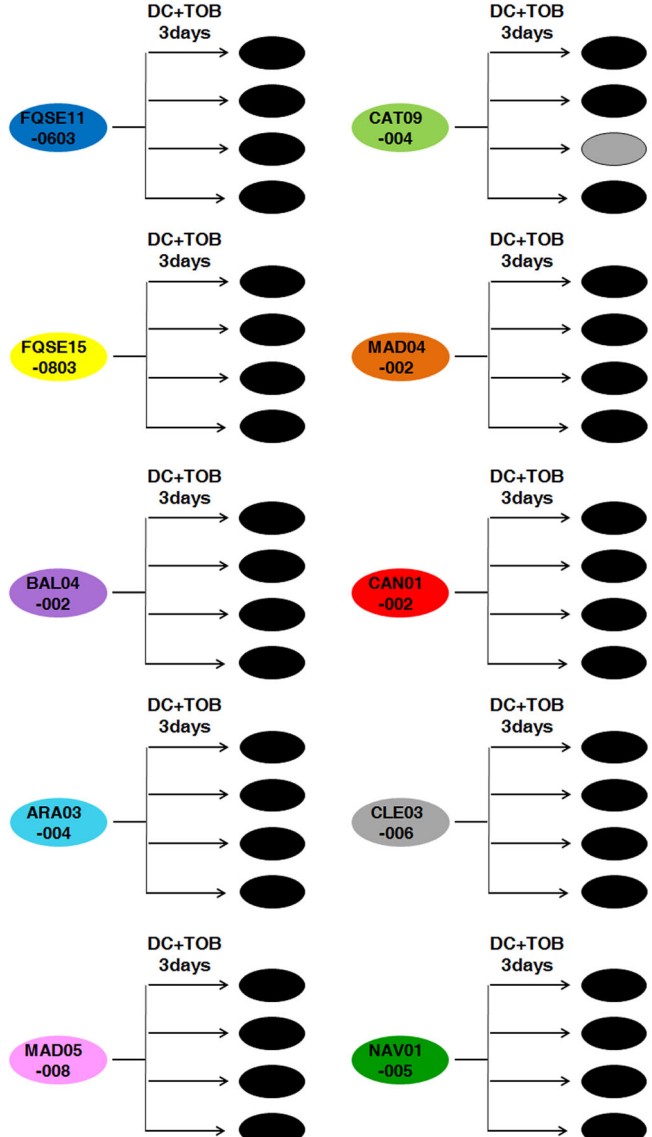

**Fig. 7 | Diagram showing the efficacy of the combination of dequalinium chloride with tobramycin for driving different clinical strains of *P. aeruginosa* to extinction.** Short-term evolution of ten clinical strains (see Supplementary Table 6), four replicate populations of each parental strain (40 populations), was performed during three days in the presence of the DC-tobramycin combination. Growth of the 80 control populations was confirmed in the presence of each single drug, at the concentrations present in the drugs combinations. Populations extinct at the end of the ALE experiment are represented in black and surviving populations are colored in gray. Most of the populations (39 out of 40) submitted to short-term ALE in the presence of the combination were extinct. These results indicate that transient CS may improve combinatory therapy and that the DC-tobramycin combination could be a very efficient evolutionary strategy to treat *P. aeruginosa* infections, including those containing tobramycin-resistant and high-risk epidemic clones.

To further confirm that transient CS is the only mechanism responsible for the extinction observed in the clinical isolates analyzed, we tested the efficacy of DC in combination with ceftazidime, an antibiotic for which DC does not induce CS (Fig. 2; Table 1) using ALE assays. We submitted 4 replicate populations of each different genetic background to the combination DC-ceftazidime (40 populations) during three days. In parallel, we grew 40 populations in DC or ceftazidime, using the same concentrations present in the drug pair, and 40 populations in the absence of any drug (120 control populations)

("Methods"). All the populations submitted to short-term ALE in the presence of the combination DC-ceftazidime grew, confirming that transient CS to tobramycin mediated by DC is the only event responsible for the extinction of the clinical strains analyzed.

Overall, these results indicate that the pair DC-tobramycin is effective in driving to extinction clinical strains presenting different genomic backgrounds, different mutational resistomes (including tobramycin-resistant mutants) and some of them belonging to high-risk epidemic clones (ST111 and ST244).

## Discussion

The existing crisis of AR requires the design of novel therapeutic strategies that improve the efficacy of available antibiotics[76]. One trade-off of AR evolution that could allow the design of efficient therapeutic strategies is CS. However, there are barriers that hinder the implementation of these CS-based treatment strategies. One of the most important is the lack of conservation of CS in different genetic backgrounds, including pre-existing antibiotic-resistant mutants, which cause infections[9]. Supporting this statement, the conclusion of a recent analysis of around 450,000 susceptibility tests, spanning a 4-year period at different University of Pittsburgh's hospitals, is that therapeutic strategies based on alternating antibiotics may be mostly ineffective due to differences in patterns of CS[77]. Therefore, the identification of robust CS patterns is critical for the application of CS-based treatment strategies.

Phenotypic convergence towards CS may be the consequence of parallel evolution[50] or the result of different genetic events[28]. Concerning the second case, we have recently identified a robust CS to tobramycin in different antibiotic-resistant mutants and clinical strains of *P. aeruginosa* presenting different genomic backgrounds, associated with the use of ciprofloxacin, caused by the selection of mutations in different genes depending on the genetic background[28,29]. However, although this phenotypic convergence made possible the design of therapeutic strategies to drive the analyzed resistant mutants to extinction, the strength of CS to tobramycin was higher when mutations in *nfxB*, and not in *gyrAB* or *mexS*, were acquired. Therefore, in this work, we tried to "circumvent" constrains imposed by historical contingency and specifically, and temporally, canalize CS by inducing the *nfxB*-mediated ciprofloxacin resistance (and tobramycin hypersusceptibility) mechanism, using DC. This approach has two benefits. One is to reduce the effect of historical contingency that makes difficult the emergence of robust, clinically exploitable, CS patterns, by funneling bacterial adaptation through the induction of a single, well-defined, mechanism of AR. The other is to achieve CS without the risky selection of antibiotic-resistant mutants.

Ciprofloxacin resistance associated with the overproduction of the MexCD-OprJ efflux pump may occur by the acquisition of loss-of-function mutations in the gene that encodes its local repressor, *nfxB*[25,26,28]. In addition, the overexpression of *mexCD-oprJ* may occur due to the presence of inducer compounds[13,18,19], which may affect the regulatory capacity of NfxB on the expression of this operon[14,15]. In particular, DC directly binds the transcriptional repressor of the TetR family RamR, which regulates the expression of the operon encoding the efflux pump AcrAB-TolC, reducing its DNA-binding affinity[78]. Similarly, DC may also bind and inhibit the regulatory capacity of NfxB leading to both, ciprofloxacin resistance and CS to tobramycin, phenotypes associated with loss-of-function mutations in *nfxB*[20,28]. In fact, the *nfxB177* mutant is the only genetic background that did not present variations in the levels of resistance to ciprofloxacin or aminoglycosides (i.e., tobramycin or amikacin) in the presence of DC, further supporting that the effect of DC is specific and principally falls on NfxB.

DC is an antiseptic and disinfectant frequently used in clinics that presents many useful therapeutic properties[62–71]. Further, and although we have previously described its capacity to reduce the efficacy of ciprofloxacin through transient AR[13], here we demonstrate

that it is able to transiently induce CS to tobramycin, avoiding the acquisition of inherited ciprofloxacin resistance (see Conceptual Fig. 1). Moreover, we show that its combination with tobramycin is effective in driving different pre-existing resistant mutants of *P. aeruginosa*, as well as clinical strains (including tobramycin-resistant and high-risk epidemic clones), to extinction. Therefore, it would be very interesting to analyze if a combined formulation of tobramycin, frequently inhaled during *P. aeruginosa* therapy[59], with DC could be effective for the treatment of pulmonary infections caused by this bacterium. In this line, an inhaled formulation of curcumin-DC for pulmonary administration of encapsulated curcumin molecules with anticancer capacities has been described[79]. However, deeper studies that determine the concentrations of DC that could be safely applied by inhalation, alone or in combination with other drugs (i.e., tobramycin), are still required.

Overall, we show that the concept of CS coined in the 1950s, understood as susceptibility to a drug associated with the acquisition of resistance mutations to another[30], and deeply studied by many scientists[27,31,34–43], is even broader, including cases where susceptibility to a second drug is neither stable nor inherited, but transient. In fact, our results constitute a proof of concept supporting the hypothesis that it is possible to exploit transient and robust CS phenotypes associated with the induction of AR. Thinking about the clinical exploitation of CS, inducing a susceptible state of transient AR is less risky than forcing the selection of stable AR. Further, the latter does not always lead to a conserved pattern of CS in different genetic backgrounds, while our approach would allow inducing the same molecular mechanism of CS, regardless of the genetic background.

Overall, our results support that the identification of new compounds able to induce CS patterns might be valuable for the design of evolution-based strategies to tackle antibiotic-resistant infections. In addition, we suggest that looking for other inducible CS mechanisms in this and in other bacterial species could constitute a relevant advance in the rational design of therapeutic approaches to manage bacterial infections.

## Methods
This research complies with all relevant ethical regulations.

### Growth conditions and antibiotic susceptibility assays
Unless stated otherwise, *P. aeruginosa* PA14, PA14-derived isogenic mutants (4 single mutants and 2 multiple mutants and ten clinical isolates of *P. aeruginosa* from different Spanish hospitals (Supplementary Table 6)[29], were grown in glass tubes in Lysogeny Broth (LB) (Lenox, Pronadisa) at 37 °C with shaking at 250 rpm. MICs of ciprofloxacin, tobramycin, amikacin, gentamycin, aztreonam, ceftazidime, imipenem, piperacillin, fosfomycin, tetracycline and erythromycin or colistin and polimixin B were determined at 37 °C, in Mueller Hinton (MH) agar or MHII agar respectively, using E-test strips (MIC Test Strip, Liofilchem®) in the presence or absence of 10 μg/ml of DC, as previously described[13], using bacteria previously grown in 1 ml of LB supplemented or not with 10 μg/ml of DC.

### Alternation of dequalinium chloride with tobramycin and combination of dequalinium chloride with tobramycin or ceftazidime using short-term adaptive laboratory evolution experiments in isogenic antibiotic-resistant mutants of *P. aeruginosa* PA14
For the alternated strategy, short-term ALE experiments in the presence of DC and tobramycin were performed for 4 days, at 37 °C and 250 rpm, using 28 replicate populations belonging to PA14 wild-type strain and 6 different mutational backgrounds (*parR87*, *orfN50*, *nfxB177*, *mexZ43*, MDR6, and MDR12), diluting (1/100) cultures in fresh LB every day and starting from glycerol stocks. Assays

consisted in one day of ALE in the presence of 10 μg/ml of DC and 3 days of ALE in the presence of tobramycin, at the concentration that hinders—but allows—the growth of each *P. aeruginosa* mutational background under these culture conditions: 0.5 μg/ml for *nfxB177*, 0.75 μg/ml for PA14, 1.5 μg/ml for MDR6 and *mexZ43*, 2 μg/ml for *orfN50*, 2.5 μg/ml for *parR87*, and 12 μg/ml for MDR12. The 28 challenged populations (DC followed by tobramycin) and the 84 control populations (28 populations constantly grown in 10 μg/ml of DC, tobramycin, or absence of any compound) were grown during 4 days.

For the combined strategies, four replicate populations from PA14 wild-type strain and 6 different mutational backgrounds were grown, from glycerol stocks. Every day, during three days, the cultures were diluted (1/100) in fresh LB medium containing a combination of DC-tobramycin (28 populations), a combination of DC-ceftazidime (28 populations), each single compound (84 control populations) or absence of compound (28 control populations). The concentration of drugs in the pairs was the same that those used individually. 10 μg/ml of DC were added for each genetic background and tobramycin or ceftazidime were added at the concentration that hinders—but allows—the growth of each *P. aeruginosa* mutational background under these culture conditions, which are described above and in ref. [50]. Extinction of the populations was determined by plating out final cultures on LB agar to look for viable cells.

### Combination of dequalinium chloride with tobramycin or ceftazidime using short-term adaptive laboratory evolution experiments in clinical strains of *P. aeruginosa*
Four replicate populations from ten different clinical isolates of *P. aeruginosa* (Supplementary Table 6) were grown, from glycerol stocks. Every day, during three days, the cultures were diluted (1/100) in fresh LB medium containing a combination of DC-tobramycin (40 populations), a combination of DC-ceftazidime (40 populations), each single compound (120 control populations) and absence of any drug (40 control populations). The concentration of drugs in the pairs was the same that those used individually. 10 μg/ml of DC were added for each clinical isolate and tobramycin or ceftazidime was added at the concentration that hinders—but allows—the growth of each *P. aeruginosa* mutational background under these culture conditions. In the case of tobramycin: 0.75 μg/ml for BAL04-002 and CAT09-004, 1.5 μg/ml for FQSE110603, FQSE150803, and CAN01-002, 2 μg/ml for ARA03-004 and MAD05-008, 3 μg/ml for MAD04-002 and CLE03-006 or 6 μg/ml for NAV01-005. In the case of ceftazidime: 0.75 μg/ml for NAV01-005, 1 μg/ml for BAL04-002, CAT09-004, FQSE110603 and CAN01-002, 2 μg/ml for CLE03-006, 2.5 μg/ml for MAD05-008 and FQSE150803, 3 μg/ml for MAD04-002 or 8 μg/ml for ARA03-004. Extinction of the populations was determined by plating out final cultures on LB agar to look for viable cells.

### Checkerboard analysis
Seven standard checkerboard broth microdilution assays were performed with PA14 wild-type strain and 6 different mutants (*parR87*, *orfN50*, *nfxB177*, *mexZ43*, MDR6, and MDR12) using serially diluted concentrations of DC, tobramycin and a no-drug control, in 96-U-well plates. For that, 90 μl of MH medium with DC (4 to 40 μg/ml) or tobramycin (0.0625 to 4 μg/ml) were added to each well of 96-U-well plates for the analysis of each mutational background with exception of MDR12, for which tobramycin concentrations were higher (1 to 64 μg/ml). 10 μl of cells were inoculated into each well to a final $OD_{600nm}$ of 0.01. Bacteria were grown at 37 °C for 48 h without shaking. The fraction inhibitory concentration (FIC) of DC and tobramycin was calculated as the MIC of the combination of DC with tobramycin divided by the MIC of each of the compounds alone. FIC index resulted from the addition of the FICs of both compounds. A FIC index value of <0.5 or >4 was considered to indicate synergy or antagonism, respectively[80].

**Reporting summary**

Further information on research design is available in the Nature Portfolio Reporting Summary linked to this article.

## Data availability

All data needed to evaluate the conclusions of this work are present in the main document and/or the Supplementary Materials.

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

## Acknowledgements

Thanks are given to our colleague and friend Álvaro San Millan (Centro Nacional de Biotecnología, CNB-CSIC) for carefully reading the

manuscript and providing useful comments for its improving, and to Antonio Oliver (Servicio de Microbiología, Hospital Universitario Son Espases from Palma de Mallorca) for providing us with the clinical strains of *P. aeruginosa*. Thanks are also given to both reviewers for their suggestions during the peer review, which substantially improved this work. This work was supported by MCIN/AEI/10.13039/501100011033 - grants PID2020-113521RB-I00 and the "Severo Ochoa" Programme for Centres of Excellence in R&D (SEV-2017-0712).

## Author contributions

S.H.A. participated in the design of the study and performed experimental work. P.L. participated performing experimental work. J.L.M. participated in the design of the study. All authors participated in writing the manuscript and approved the submitted version.

## Competing interests

The authors declare no competing interests.
