## [Peer review file · Nature Communications]

REVIEWER COMMENTS

Reviewer #1 (Remarks to the Author):

In the manuscript "Tackling antibiotic resistance by inducing transient and robust collateral sensitivity" Hernando-Amado et al. investigated the presence and exploitation of induced and transient collateral sensitivity to antibiotics. As the high rate of spread of antibiotic resistant bacterial pathogens is a problem of ever-increasing importance, new approaches to fight against them are of great necessity.

In the introduction the authors state that the existence of different genetic backgrounds often limits the clinical exploitation of collateral sensitivity (CS) as there are a very limited number of cases in which a robust pattern of CS emerges when a drug is used. They offer a possible solution to this problem by inducing a resistance mechanism (efflux pump- MexCD-OprJ) by dequalinium chloride (DC) in different, even unrelated clinical *P. aeruginosa* isolates. Inducing the MexCD-OprJ overproduction by DC provides moderate resistance to quinolones and at the same time collateral sensitivity/hypersensitivity to different aminoglycosides even in strains previously being resistant to tobramycin which is indeed very interesting. They also highlight and demonstrate its other important aspect that the capacity of DC to induce resistance to ciprofloxacin and CS to tobramycin in the same time is only temporary, rapidly reversing when its presence ceases. In addition, they also show that transient CS may drive antibiotic-resistant mutants and clinical strains of *P. aeruginosa* to extinction. Overall, the study raises very important new aspects of evolution-based treatment strategies to tackle antibiotic-resistance by applying compounds to induce robust, homogenous and transient collateral sensitivity. However, I miss some very important control experiments that would substantially strengthen their findings that induced and transient CS are responsible and can be used to drive antibiotic-resistant mutants and clinical strains of *P. aeruginosa* to extinction.

Major points:

What is the effect of the antiseptic DC on the clearance (killing) efficacy of the tobramycin? Is it CS dependent or not? Do you see different clearance interactions on the nfxB177 (where the DC did not induce CS to tobramycin).

It would be important to test the outcome of the combined ALE assay on the nfxB177 strain as well (where the DC did not induce CS). If the CS is purely responsible for the extinction of the resistant strains then the nfxB177 populations would survive the combination treatment. However, if the DC strongly synergizes the clearance efficacy of the tobramycin, the combination treatment will also lead to extinction of the nfxB177 populations as well.

To further strengthen the result that transient CS is responsible to drive antibiotic-resistant mutants and clinical strains of *P. aeruginosa* to extinction. It would be also important to test the outcome of the combined ALE assay using a different antibiotic than TOB, for example an antibiotic against the DC does not induce CS (like ceftazidime and/or piperacillin).

Minor points:

I would argue about the statement of the authors that exploitation of CS, as explored so far, always requires the evolution of resistance to one of the drugs which is risky / undesirable if these resistant strains can not be efficiently eradicated by the second antibiotic of the pair. For example another possible application of CS is when collateral sensitive drug pairs are given together to limit or substantially slow down the resistance evolution to either both or one of the drugs of the combination (Lazar et al, 2018, Nature Microbiology).

Were the cultures diluted each day during the alternated ALE assay? It is not clear from the relevant Method section.

Was the DC used in sublethal dosage? What is the MIC of DC on the different *P.aeruginosa* isolates?

Is the exception genotype (nfxB177) hypersensitive or resistant to tobramycin compared to its ancestor?

Was the concentration of tobramycin the same in the combination and single treatments during the ALE assay?

Reviewer #2 (Remarks to the Author):

The authors present an interesting property of the compound DC, leading to transient collateral sensitivity against TOB. They evaluate in populations derived in the lab through experimental evolution,

as well as clinical isolates. The experiments are short and it is unclear whether there is selection under the use of DC or whether there is enough time considered for compensatory mutations to emerge. While the results are interesting, the presented data is insufficient for the conclusions of the authors, and more in depth evolution experiments and genomic data is necessary to validate their claims. There is a clear potential for the suggested use of DC, but the evidence and data presented here is still superficial and incomplete

Main concerns:

- The authors argue that in the remaining populations after experimental evolution did not evolve resistance to CIP and CS to TOB. Yet, the time for all the experiments were never longer than 3 days? How many generations is this? Is this enough for a de novo mutation to emerge? As presented now, it is unclear whether compensatory mutations may emerge in longer time scales, or whether DC selects for other type of mutations. This is also relevant because the authors suggest that DC can be used inhaled in pulmonary infections; yet such treatments require longer durations.
- It seems the authors knew about the very short induction of CS and CIP resistance using DC. Then, why test its effect in a sequential manner?

Minor comments:

- Too heavy use of acronyms throughout the manuscript. It is detrimental for the natural reading flow.
- In the introduction, the authors say: “ The exploitation of CS, as explored so far, requires the selection of antibiotic-resistant mutants to a first drug. This will be a drawback if these were not efficiently eradicated by the second antibiotic of the pair.” This is a drawback, regardless of CS. Perhaps, the more important drawback is that selection resulting in resistance and collateral sensitivity simultaneously may not occur.
- In the last paragraph of the introduction something is missing in the phrase: Among the possible inducers, we chose dequalinium chloride (DC) presents many pharmacological and therapeutic benefits, including its antiseptic and disinfectant capacities.
- Is there no selection by DC alone?
- Figure 2 though relatively simple, is not easy to interpret. The reader needs to make a double calculation. For clarity, I would suggest including two heatmaps rather than the foldchange relative to the the absence of DC. Perhaps a heatmap with the foldchange relative to PA14 and another in the presence of DC but still relative to PA14 in the absence of DC would be more clear. Especially because in the absence of DC most of the genetic backgrounds are resistant to TOB. This would make it clear that there is a change in resistance against CIP in both cases (presence and absence) and a contrasting

change in TOB; resistance increase in the absence of DC and sensitivity increase in the presence of DC. This would also eliminate the need for figure 3, which in my opinion is not only not a clear graph but is also not adding much, but I might be missing something.

- There is no need to mention the foldchanges in the text of the results as often as the authors do. It is distracting and unnecessary, especially since this is provided in Table S2. In fact, moving Table S2 to the main text might improve the flow of the text.
- The authors mention: "These results also indicate that, while stable CS to tobramycin in some mutational backgrounds would be linked to important levels of resistance to ciprofloxacin, depending on the possible mutations they may acquire, the induction of transient and fairly mild nfxB-mediated resistance to ciprofloxacin in all of them by the use of DC renders an improved CS to tobramycin (Figure 3)." However, there is only one genetic background consistently leading to CS, nfxB.

Reviewer #1 (Remarks to the Author):

In the manuscript "Tackling antibiotic resistance by inducing transient and robust collateral sensitivity" Hernando-Amado et al. investigated the presence and exploitation of induced and transient collateral sensitivity to antibiotics. As the high rate of spread of antibiotic resistant bacterial pathogens is a problem of ever-increasing importance, new approaches to fight against them are of great necessity. In the introduction the authors state that the existence of different genetic backgrounds often limits the clinical exploitation of collateral sensitivity (CS) as there are a very limited number of cases in which a robust pattern of CS emerges when a drug is used. They offer a possible solution to this problem by inducing a resistance mechanism (efflux pump- MexCD-OprJ) by dequalinium chloride (DC) in different, even unrelated clinical *P. aeruginosa* isolates. Inducing the MexCD-OprJ overproduction by DC provides moderate resistance to quinolones and at the same time collateral sensitivity/hypersensitivity to different aminoglycosides even in strains previously being resistant to tobramycin which is indeed very interesting. They also highlight and demonstrate its other important aspect that the capacity of DC to induce resistance to ciprofloxacin and CS to tobramycin in the same time is only temporary, rapidly reversing when its presence ceases. In addition, they also show that transient CS may drive antibiotic-resistant mutants and clinical strains of *P. aeruginosa* to extinction. Overall, the study raises very important new aspects of evolution-based treatment strategies to tackle antibiotic-resistance by applying compounds to induce robust, homogenous and transient collateral sensitivity. However, I miss some very important control experiments that would substantially strengthen their findings that induced and transient CS are responsible and can be used to drive antibiotic-resistant mutants and clinical strains of *P. aeruginosa* to extinction.

We really appreciate the opinion of reviewer 1 about our work. He/she not only appreciated the strategy we propose and the fundamental basis of this work but he/she suggested a series of experiments (see below) that have clearly improved our article.

Major points:

What is the effect of the antiseptic DC on the clearance (killing) efficacy of the tobramycin? Is it CS dependent or not? Do you see different clearance interactions on the *nfxB177* (where the DC did not induce CS to tobramycin).

We appreciate the suggestions of reviewer 1 regarding the *nfxB177* mutant, since a deeper analysis of this mutant has strengthened our results: transient CS is responsible to drive different strains of *P. aeruginosa* to extinction. As the reviewer 1 suggested, we added to our checkerboard analysis (initially performed in PA14, *parR87*, *orfN50*, *mexZ43*, MDR6 and MDR12) the *nfxB177* mutant to complete the synergistic analysis between DC and tobramycin performed in the different genetic backgrounds. We did not observe synergy between DC and tobramycin with this mutant (FIC Index > 0.5; Supplementary Table 2). In fact, when we look at changes in MIC to tobramycin in the different genetic backgrounds in the presence and absence of DC, *nfxB177* is the only one that does not show a reduction in tobramycin MIC. Further, ALE experiments suggested by reviewer 1 (see below) on the *nfxB177* strain indicate that CS is purely responsible for the extinction of the resistant strains. These results are now included in the article.

It would be important to test the outcome of the combined ALE assay on the *nfxB177* strain as well (where the DC did not induce CS). If the CS is purely responsible for the extinction of the resistant strains then the *nfxB177* populations would survive the combination treatment. However, if the DC strongly synergizes the clearance efficacy of the tobramycin, the combination treatment will also lead to extinction of the *nfxB177* populations as well.

We would like to thank Reviewer 1 for suggesting this experiment that is indeed a nice control for verifying our hypothesis. As proposed, we performed ALE assays with this mutant to analyze the efficacy of the alternated and combined strategies. In both cases, the different replicate populations (four/treatment) of this mutant grew at the end of the ALE assay indicating that, as stated by the referee, "CS is purely responsible for the extinction of the resistant strains"; in other words, transient *nfxB*-mediated CS to tobramycin mediated by DC is responsible for the extinction of the resistant mutants because, in referee's words, "the *nfxB177* populations survive the combination treatment". Further, we have confirmed that there are no other molecular mechanisms, beyond transient CS, responsible for the extinction observed in the antibiotic-resistant mutants, following a nice suggestion of reviewer 1 (see below). These results are now presented in the article.

To further strengthen the result that transient CS is responsible to drive antibiotic-resistant mutants and clinical strains of *P. aeruginosa* to extinction, it would be also important to test the outcome of the combined ALE assay using a different antibiotic than TOB, for example an antibiotic against the DC does not induce CS (like ceftazidime and/or piperacillin).

We thank the referee for this smart idea that constitutes an easy-to-perform, but strong control. We used ceftazidime since, as she/he mentioned, we did not observed changes in MIC to this antibiotic in the presence of DC in any of the genetic backgrounds analyzed (PA14, *parR87*, *orfN50*, *nfxB177*, *mexZ43*, MDR6 and MDR12; Figure 2, Table 1) and, after 3 days of combined treatment, we did not find extinction of any of the replicate populations analyzed. In addition, we also performed this experiment with the ten clinical strains included in the work. All the populations grew after 3 days in the presence of DC-ceftazidime. These results suggest that there is no other molecular mechanism, beyond transient CS, causing the extinction of the different strains analyzed. Further, it supports that the effect is specific for tobramycin. This novel information is now presented in the article.

Minor points:

I would argue about the statement of the authors that exploitation of CS, as explored so far, always requires the evolution of resistance to one of the drugs which is risky / undesirable if these resistant strains cannot be efficiently eradicated by the second antibiotic of the pair. For example another possible application of CS is when collateral sensitive drug pairs are given together to limit or substantially slow down the resistance evolution to either both or one of the drugs of the combination (Lazar et al, 2018, Nature Microbiology).

Reviewer 1 is correct and we were actually aware of this seminal work about the potential of antimicrobial peptides when used in combination with other drugs. Since it is true that CS can also be exploited when an antibiotic (so far described antimicrobial peptides) slows down the evolution of resistance, we have highlighted this possibility in the text, maintaining that there is a risk of selecting resistant mutants if they were not efficiently eradicated by the second drug of the pair.

Were the cultures diluted each day during the alternated ALE assay? It is not clear from the relevant Method section.

We have clarified the text; cultures were diluted (1/100) every day, as we did during combined ALE assays.

Was the DC used in sublethal dosage? What is the MIC of DC on the different *P.aeruginosa* isolates?

We used a sublethal concentration of DC (10 µg/ml), the lowest that allowed us to observe the induction of CS to tobramycin in all the strains. On the other hand, it was not possible to determine the MIC to DC due to the solubility limit of said compound, growing all the genetic backgrounds analyzed at least at 40 µg/ml.

Is the exception genotype (*nfxB177*) hypersensitive or resistant to tobramycin compared to its ancestor?

This mutant presents a loss-of-function mutation which leads to resistance to ciprofloxacin (MIC 2) and increased susceptibility to tobramycin (MIC 0.5) compared with the PA14 parental strain (MICs 0.064 and 1, respectively). In fact, we have also checked other *nfxB* mutant from our lab (Linares et al, 2005., Journal of Bacteriol), but in a PAO1 context, and its loss-of-function also causes resistance to ciprofloxacin (MIC 1) and increased susceptibility to tobramycin (MIC 1) as compared with the PAO1 parental strain (MICs 0.064 and 2, respectively). Answering reviewer 1, the *nfxB177* mutant included in this work is a loss-of-function mutant (resistant to ciprofloxacin and susceptible to tobramycin), something we have now indicated in the text.

Was the concentration of tobramycin the same in the combination and single treatments during the ALE assay?

We thank reviewer 1 for having noticed that we had not indicated in the materials section, although we did in the results, that the concentration of the drugs present in the pairs were the same that we used in single treatments. We have indicated it also in the materials section.

Reviewer #2 (Remarks to the Author):

The authors present an interesting property of the compound DC, leading to transient collateral sensitivity against TOB. They evaluate in populations derived in the lab through experimental evolution, as well as clinical isolates. The experiments are short and it is unclear whether there is selection under the use of DC or whether there is enough time considered for compensatory mutations to emerge. While the results are

interesting, the presented data is insufficient for the conclusions of the authors, and more in depth evolution experiments and genomic data is necessary to validate their claims. There is a clear potential for the suggested use of DC, but the evidence and data presented here is still superficial and incomplete.

We are glad to know that reviewer 2 finds our work interesting and that he/she believes in the potential of DC. Regarding his/her doubts about the ALE assays performed, we would like to state that we have used this methodology (short-term adaptive laboratory evolution assays) in different works, in search for robust patterns of AR and CS, finding that three days of ALE (around 20 generations) is enough to select for AR mutations (please, see Hernando-Amado et al 2020., SciAdvances, Hernando-Amado et al 2022., PNAS and Hernando-Amado et al 2022 MicrobSpectrum). In fact, resistance to ciprofloxacin is selected, via mutations in *nfxB* (causing the same effect as DC: *mexCD-oprJ* overexpression), in only three days, both in laboratory and clinical strains (please, see Hernando-Amado et al 2022., PNAS and Hernando-Amado et al 2022 MicrobSpectrum). Regarding compensatory mutations, in case that AR was selected and it had an associated cost, it would be expected that it would take longer to occur (around 400 generations in: Hernando-Amado et al 2022., MolBiolEvol and Dunai et al 2019., eLife). However, this work does not propose a long treatment (in which there will be risk of selection of resistance mutations and in which compensatory mutations could be acquired) but a short one, which, in just three days, is effective causing extinction of the mutants and clinical strains analyzed. Anyway, following concerns of reviewer 2 (see below), we have analyzed if resistance to other antibiotics (different to ciprofloxacin and tobramycin) could have emerged in our populations, finding no differences in resistance levels and, in addition, we have reached 100 generations evolving our populations in presence of DC finding that it does not lead to an increased ciprofloxacin or tobramycin resistance and that CS to tobramycin is still inducible. All this information is now included in the article.

Main concerns:

- The authors argue that in the remaining populations after experimental evolution did not evolve resistance to CIP and CS to TOB. Yet, the time for all the experiments were never longer than 3 days? How many generations is this? Is this enough for a de novo mutation to emerge? As presented now, it is unclear whether compensatory mutations may emerge in longer time scales, or whether DC selects for other type of mutations. This is also relevant because the authors suggest that DC can be used inhaled in pulmonary infections; yet such treatments require longer durations.

We hope we have answered these questions from reviewer 2 in the previous paragraph. Indeed, we not only propose a short therapeutic strategy based on previous results from our laboratory that show the efficacy of short treatments (please, see Hernando-Amado et al 2020., SciAdvances, Hernando-Amado et al 2022., PNAS and Hernando-Amado et al 2022 MicrobSpectrum) but rather, there is robust evidence that short antibiotic therapies can be as effective as longer ones for the treatment of several infections. In fact, a paper recently published focusing on the extent to which the field of infectious diseases can overcome the inertia of past, historical, medical decisions, authors indicate: “more than 120 modern, randomized controlled trials have established that shorter durations of therapy are equally effective for many infections” (please, see Davar et al

2023., *OpenForumInfectDis* and Lee RA et al 2021., *AnnInterMed*). These findings are now discussed and the corresponding references have been added. Having said that, and although in this work we do not propose a long treatment but a short one, and following concerns of reviewer 2, we have analyzed if resistance to other antibiotics (different to ciprofloxacin and tobramycin) could have emerged in the populations after the short use of DC, finding no differences in resistance levels and, in addition, and although this is far from the strategy we propose, we reached 100 generations evolving our populations in presence of DC finding that it does not select for ciprofloxacin or tobramycin resistance and that it is still possible to induce CS to tobramycin in the different resistant mutants using DC, as we observed after a short treatment (4 days of ALE). We hope that these reasoning together with the new experiments carried out resolve the doubts raised by reviewer 2.

- It seems the authors knew about the very short induction of CS and CIP resistance using DC. Then, why test its effect in a sequential manner?

As reviewer 2 said, we knew about the rapid decline of DC induction from experiments previously carried out in the laboratory (Laborda et al 2019., *AAC*) in which we washed the cultures and measured the "memory" of the induction. However, we had not previously done an ALE assay testing the durability of the phenotype associated with the use of this compound (CS to tobramycin) which, on the other hand, is a phenotype that we had not described before. Therefore, it was of interest to test it although, as it occurred when alternating or combining ciprofloxacin and tobramycin in pre-existing mutants and in clinical strains (please, see Hernando-Amado et al 2022., *PNAS* and Hernando-Amado et al 2022 *MicrobSpectrum*), the treatment that efficiently works is the combined one. Anyway, we have performed a new a series of ALE experiments suggested by reviewer 1 that have clearly strengthened our ALE results and we hope that reviewer 2 finds this new information relevant. This information has been included in the new version of the article.

Minor comments:

- Too heavy use of acronyms throughout the manuscript. It is detrimental for the natural reading flow.

We appreciate the suggestion of reviewer 2 as we thought that we had tried to use acronyms in the regular way they are used: when a combination of words is repeated more than three times in the text. However, we found one acronym (MDR) that was repeated just twice and it has been eliminated. Nevertheless, if the referee or the editor believe that we have abused using acronyms, we are open for suggestions on which ones would be good to remove.

- In the introduction, the authors say: “ The exploitation of CS, as explored so far, requires the selection of antibiotic-resistant mutants to a first drug. This will be a drawback if these were not efficiently eradicated by the second antibiotic of the pair.” This is a drawback, regardless of CS. Perhaps, the more important drawback is that selection resulting in resistance and collateral sensitivity simultaneously may not occur.

At the request of reviewer 2 and reviewer 1, we have modified this sentence. We believe that it is now more accurate.

- In the last paragraph of the introduction something is missing in the phrase: Among the possible inducers, we chose dequalinium chloride (DC) presents many pharmacological and therapeutic benefits, including its antiseptic and disinfectant capacities.

We thank reviewer 2 for finding this typo, we have fixed it.

- Is there no selection by DC alone?

As we explained above, with the experimental approach used, DC did not select for resistance to the antibiotics tested after 20 generations. Now, we have expanded the analysis for a longer time and we did not find an increase in ciprofloxacin or tobramycin MICs after 100 generations of evolution in presence of DC alone. More importantly, we found that the capacity of DC to induce transient CS to tobramycin is preserved. This information is now included in the article.

- Figure 2 though relatively simple, is not easy to interpret. The reader needs to make a double calculation. For clarity, I would suggest including two heatmaps rather than the foldchange relative to the the absence of DC. Perhaps a heatmap with the foldchange relative to PA14 and another in the presence of DC but still relative to PA14 in the absence of DC would be more clear. Especially because in the absence of DC most of the genetic backgrounds are resistant to TOB. This would make it clear that there is a change in resistance against CIP in both cases (presence and absence) and a contrasting change in TOB; resistance increase in the absence of DC and sensitivity increase in the presence of DC. This would also eliminate the need for figure 3, which in my opinion is not only not a clear graph but is also not adding much, but I might be missing something.

We appreciate reviewer 2's suggestion regarding the possibility of representing the MIC values of each genetic context in the presence of DC compared to that of PA14. However, we want to show how DC changes MICs in each specific genetic background acting on a specific molecular mechanism. Therefore, showing just how much DC varies MICs in a mutant respect to PA14 would be misleading. For this reason, we have kept the figure as it was but, trying to clarify it, we have added the specific values to the graph (not only the color intensity) to avoid having to resort to supplementary tables. In addition, as reviewer 2 suggested (see below), we have moved Table S2 to the main text (Table 1). We believe that these two changes suggested by reviewer 2 clarify the paper and we hope that reviewer 2 finds this modification positive. If she/he believes that after having changed Fig 2 we should keep this information in the Supplementary file, we will be happy to do it.

- There is no need to mention the foldchanges in the text of the results as often as the authors do. It is distracting and unnecessary, especially since this is provided in Table S2. In fact, moving Table S2 to the main text might improve the flow of the text.

We would like to maintain the specific fold changes cited in the text, since we think it helps to understand the results, but, as reviewer 2 suggested, we have moved Table S2 to the main text (Table 1).

- The authors mention: "These results also indicate that, while stable CS to tobramycin in some mutational backgrounds would be linked to important levels of resistance to ciprofloxacin, depending on the possible mutations they may acquire, the induction of transient and fairly mild nfxB-mediated resistance to ciprofloxacin in all of them by the use of DC renders an improved CS to tobramycin (Figure 3)." However, there is only one genetic background consistently leading to CS, nfxB.

Opposite to what reviewer 2 states, *nfxB177* is the only genetic background that does not lead to transient tobramycin CS in presence of DC. Obviously if a reader (reviewer 2 in this case) gets a wrong conclusion, the problem is not with the reader, but with the way of presenting the information. It is then clear that the Figure 3 legend was misleading, causing the misunderstanding of the information. Consequently, the Figure legend has been completely re-written to make it understandable and clearer. We hope that this new legend more properly explains our findings.

REVIEWER COMMENTS

Reviewer #1 (Remarks to the Author):

For the record, my assessment was then that "The study raises very important new aspects of evolution-based treatment strategies to tackle antibiotic-resistance by applying compounds to induce robust, homogenous and transient collateral sensitivity."

However, I missed some important control experiments. In this revised version, the authors have substantially strengthened the conclusion of the paper by performing these new experiments. The authors also extended the method sections with important details of the ALE experiments.

Viktoria Lázár

Reviewer #2 (Remarks to the Author):

The authors have done a fantastic job including new data and making the manuscript clearer. I think the data presented in this study has a lot of potential. The authors have conducted numerous experiments and are trying to present evidence about a promising effect induced by DC in *P. aeruginosa*. Despite this giant amount of work, I still struggle with the current version of the manuscript.

- The other reviewer also wondered about the selective strength (MIC, dose, etc) of DC alone. The authors mention that the concentration used is the minimal possible to induce the CS effect due to solubility effects. But, given that DC has bactericidal effects, it is important to show how the inducible effect relates to dose, and how that affects the potential selection for resistance against it. At the very minimum the authors need to discuss what concentrations of DC are typically used therapeutically in inhaled form, and/or in vaginal use. This is important beyond the argument of the authors - however valid it is - that they here suggest a short term strategy, and even if the concentration required to induce the effect is lower than that of the regular use of DC, especially because the authors mention its potential use in that form and its clinical potential.

- The authors evaluated the DC effect on clinical strains but do not consider testing against the different antibiotics shown in figure 2. From a second look at this figure, the possibility emerges that the inducible effect could extend beyond aminoglycosides into other protein inhibitor classes, as seen in MDR6. Given the distinct genetic backgrounds of the clinical isolates, more patterns could emerge, clarifying the mechanism of action and the range of the inducible effect.

- Related to the previous point, the additional evolution experiments with ceftazidime, although conclusive, limit the generality of the results. Is the inducible effect seen in other aminoglycosides, other protein inhibitors, and other antibiotics more broadly.

As I said before the data already presented by the authors is impressive. But it remains insufficient to support their over-reaching claims. A more systematic and in depth approach needs to be considered to further support their claims, or to assert their conclusions better. Considering the already large experimentation carried by the authors, it is possible that they consider this suggestions or concerns beyond the reach of the current version of the paper. But, given the claims of their conclusion, I consider the more systematic and general approach could be appropriate.

Other minor issues:

- The readability of the paper is difficult in some sections of the manuscript. Some paragraphs include very long phrases that are hard to interpret, and other paragraphs include a list of fold changes that are "readable" already in the main figures and tables. Several parts of the manuscript could benefit from some editing that would make reading and interpreting the results easier.

REVIEWER COMMENTS

Reviewer #1 (Remarks to the Author):

For the record, my assessment was then that "The study raises very important new aspects of evolution-based treatment strategies to tackle antibiotic-resistance by applying compounds to induce robust, homogenous and transient collateral sensitivity."

However, I missed some important control experiments. In this revised version, the authors have substantially strengthened the conclusion of the paper by performing these new experiments. The authors also extended the method sections with important details of the ALE experiments.

Viktoria Lázár

We appreciate the positive opinion of Dr. Lázár and we are very grateful for the requests and suggestions she has made during the review of this article. She proposed different experiments that brought an important improvement of our work. We sincerely thank Viktória for her contribution and for her time.

Reviewer #2 (Remarks to the Author):

The authors have done a fantastic job including new data and making the manuscript clearer. I think the data presented in this study has a lot of potential. The authors have conducted numerous experiments and are trying to present evidence about a promising effect induced by DC in *P. aeruginosa*. Despite this giant amount of work, I still struggle with the current version of the manuscript.

We are glad to know that reviewer 2 considers us to have done a "fantastic job" that has required a "giant amount of work" and also that he/she agrees with us that the manuscript is now much clearer. We were surprised to know that the referee 2 suggests further experimental work, in addition to the experiments that she/he suggested in the first round of peer-review; all of them performed and included in this new version of the work. We are always glad to receive suggestions and, actually, we think that the experiments suggested in the first round of reviews were valuable and served to strengthen the paper. We wish like thanking both reviewers for that. However, on this occasion, we feel that the new experiments proposed by reviewer 2 not only would not strengthen the article, but would also lead to a loss of the focus of the work. The reasons for this statement are detailed below.

- The other reviewer also wondered about the selective strength (MIC, dose, etc) of DC alone. The authors mention that the concentration used is the minimal possible to induce the CS effect due to solubility effects. But, given that DC has bactericidal effects, it is important to show how the inducible effect relates to dose, and how that affects the potential selection for resistance against it. At the very minimum the authors need to discuss what concentrations of DC are typically used therapeutically in inhaled form, and/or in vaginal use. This is important beyond the argument of the authors - however valid it is - that they here suggest a short term strategy, and even if the concentration required to induce the effect is lower than that of the regular use of

DC, especially because the authors mention its potential use in that form and its clinical potential.

We used the lowest concentration of DC (10 µg/ml) that allowed us to observe the induction of CS to tobramycin in all the strains. We decided to do so to avoid any deleterious effect that could be related to the use of this compound and, in a new experiment suggested by him/her in the first round of peer-review, we demonstrated that no resistance to antibiotics emerged after a short/long treatment and that, in fact, CS to tobramycin was still inducible by DC after the longest treatment analyzed. As reviewer 2 proposed, we have carefully searched for information regarding the concentrations of DC typically used into the clinics as well as for information regarding the maximum tolerated dose in mice models. There are different medicinal products containing DC that are used for vaginal administration and the dose tested for the treatment of these infections range from 10 to 50 mg (Mendling W et al., 2015). There can be found creams (20 mg DC), ovules (20-50 mg DC) and tablets (10 mg DC). For DC Tablet therapy of vaginal infections, six days has been described as a safe and effective option. In addition, and as mentioned in the text, DC has also been used an inhaled formulation for pulmonary administration of encapsulated curcumin molecules with anticancer capacities (Zupancic S et al., 2014). In this case, the combination was tolerated at 3 µM. However, it was not analyzed the specific concentration of each of the compounds, in particular of DC, that could be safely inhaled alone or in combination with other drugs. Finally, some authors have studied the tolerated dose of DC in mice models. It has been described that the maximum tolerated dose is 15 mg/kg, tolerating 100% of the animals 5 doses of 11 mg/kg (every 7 days) (Gamboa-Vujicic G et al., 1992). In addition, it has been described that concentrations bellow that (2 mg/kg) present an important anticarcinoma activity and are well tolerated (Weiss MJ et al., 1987). Having into account all this information and since there are still required more studies that analyze the possible application of DC, as inhaled formulations, into the clinic, we have tempered our conclusions regarding its clinical potential.

- The authors evaluated the DC effect on clinical strains but do not consider testing against the different antibiotics shown in figure 2. From a second look at this figure, the possibility emerges that the inducible effect could extend beyond aminoglycosides into other protein inhibitor classes, as seen in MDR6. Given the distinct genetic backgrounds of the clinical isolates, more patterns could emerge, clarifying the mechanism of action and the range of the inducible effect.

Here, and in the next query, reviewer 2 requests more studies to "generalize" the results. In the case of compounds, obtained from one screening, studies regarding generalization are worth doing. However, we do not present this type of study. Rather, what we present is a hypothesis-driven study based on a very specific question. We knew that mutations in *nfxB*, leading to *mexCD-oprJ* overexpression, drive CS to tobramycin (and to other aminoglycosides) and we wanted to know if the temporal induction of the expression of this operon would also lead to the same phenotype. Because of this, we specifically chose DC as inducer. We identified this compound, DC, in a previous screening from our lab and, in this work, we validate our novel hypothesis. What we specifically analyze is the capacity of DC to induce a robust and transient tobramycin hyper-susceptible state in different genetic backgrounds (a varied set of antibiotic-resistant mutants) of *P. aeruginosa* and, in addition, we validated that this phenotype is specific for aminoglycosides. Second, and since it is known that DC may interfere with the activity of diverse proteins, we wondered if this effect was specific for aminoglycosides, and this is the reason why we tested other antibiotics. All these experiments supported our hypothesis. Induction rendered a robust CS to aminoglycosides and we found conservation in the different

genetic backgrounds analyzed. As stated in the article and discussed by other authors in the field, the weakest aspect of CS is that it is very often not conserved, making impossible to exploit it. Opposite to reviewer 2's statement, identifying a CS pattern that emerges in a single clinical isolate will not be useful, because exceptions cannot be easily transferred into clinical practice. But further, these suggested experiments will lead to a loss of the focus of the work which is: - is it possible to transiently induce CS to tobramycin in different *P. aeruginosa* strains, including antibiotic-resistant mutants, using DC? We believe that the data present in the current version of this work support that the effect of DC, as inducer of transient CS to tobramycin, is robust and that it can be exploited to drive antibiotic-resistant mutants, including clinical strains, to extinction. Finally, it is important to notice that reviewer 2 did not express any concern about this issue during first round of peer-review. We hope that he/she understands our arguments.

- Related to the previous point, the additional evolution experiments with ceftazidime, although conclusive, limit the generality of the results. Is the inducible effect seen in other aminoglycosides, other protein inhibitors, and other antibiotics more broadly.

It seems that reviewer 2 believes that CS is associated with the mechanisms of action (since he/she suggests testing inhibitors of protein synthesis) and this is not the case, at least in the present article. As we discuss in the text and as previously described by other authors in MexCD-OprJ overexpressing mutants, CS is likely due to a reduced amount of the porin OprM, which forms part of the aminoglycosides' MexXY efflux pump. In other words, CS is associated with the mechanisms of resistance, not with the mechanism of action. In addition, we fully agree with the reviewer 2 that ALE results on ceftazidime cannot be generalized to other antibiotics. Actually, the hypothesis behind these experiments was exactly this one: induction of CS by DC is specific for tobramycin (and other aminoglycosides, see Figure 2) and it does not apply for other antibiotics. In this regard, the ALE experiments performed in the presence of ceftazidime were not performed to "generalize" our results to other type of antibiotics. In fact, this experiment was a negative control, where the populations should not be extinct (DC does not induce CS to ceftazidime). This smart control, suggested by reviewer 1, in fact worked very well and it supports our hypothesis and strengthens our results. Finally, and as we discussed above, although several antibiotics were tested (Figure 2), the only ones for which the different genetic backgrounds showed a robust CS pattern were aminoglycosides. Actually, it is a generalization in which specificity matters. As above stated, making new ALE experiments for which bacteria populations are not susceptible in the presence of DC, or for antibiotics where a single isolate presents CS, is not only useless but misleading.

As I said before the data already presented by the authors is impressive. But it remains insufficient to support their over-reaching claims. A more systematic and in depth approach needs to be considered to further support their claims, or to assert their conclusions better. Considering the already large experimentation carried by the authors, it is possible that they consider this suggestions or concerns beyond the reach of the current version of the paper. But, given the claims of their conclusion, I consider the more systematic and general approach could be appropriate.

As previously said, we are glad to know that reviewer 2 appreciates the important amount of work we have performed to get an improved version of our work. We disagree with her/his statement that "a more systematic and in depth approach needs to be considered to further support their claims" because, as above discussed, generalization is unneeded. We formulated

an hypothesis and it was validated, and this was not questioned in the first round of review by reviewer 2. Having said that, and despite reviewer 2's opinion about the lack of generalization of our results, it is important to mention that several antibiotics were tested (not to generalize but to rule out non-specific effects of DC; please, see Figure 2) and we found that robust CS to other drugs is not induced by DC (Figure 2). The conclusion is that the effect is specific; that is why we believe that performing the new experiments suggested by reviewer 2 is not only unneeded but they will lead to a loss of the focus of the work.

Other minor issues:

- The readability of the paper is difficult in some sections of the manuscript. Some paragraphs include very long phrases that are hard to interpret, and other paragraphs include a list of fold changes that are "readable" already in the main figures and tables. Several parts of the manuscript could benefit from some editing that would make reading and interpreting the results easier.

We have tried to make more readable and understandable our manuscript based on the suggestion of reviewer 2. Regarding fold changes, we would like to maintain them in the text, since we think it helps to understand the results.